# Experimental Investigation of Metal-Based Calixarenes as Dispersed Catalyst Precursors for Heavy Oil Hydrocracking

Mohamed Ibrahim [1], Fahad A. Al-Zahrani [1], Francisco J. Diaz [2], Tareq Al-Attas [3], Hasan Zahir [4], Syed A. Ali [5], Mohammed Abdul Bari Siddiqui [5] and Mohammad M. Hossain [1,5,*]

1 Department of Chemical Engineering, King Fahd University of Petroleum and Minerals, Dhahran 31261, Saudi Arabia
2 Department of Chemical Engineering, Universidad del Pais Vasco, 48940 Leioa, Spain
3 Department of Chemical and Petroleum Engineering, University of Calgary, Calgary, AB T2N 1N4, Canada
4 Interdisciplinary Research Center for Renewable Energy and Power Systems (IRC-REPS), King Fahd University of Petroleum and Minerals, Dhahran 31261, Saudi Arabia
5 Interdisciplinary Research Center for Advanced Materials, King Fahd University of Petroleum and Minerals, Dhahran 31261, Saudi Arabia
* Correspondence: mhossain@kfupm.edu.sa; Tel.: +966-13-860-1478; Fax: +966-13-860-4234

**Abstract:** Slurry-phase hydrocracking utilizing metal-containing oil-soluble compounds as precursors of dispersed catalysts is an effective approach for heavy oil upgrading. We propose applying metal-based p-tert-butylcalix[6]arene (TBC[6]s) organic species as dispersed catalyst precursors to enhance catalytic hydrogenation reactions involved in the upgrading of vacuum gas oil (VGO). Co- and Ni-based TBC[6]s were synthesized and characterized by SEM-EDX, ICP, XRD, and FT-IR. The thermogravimetric and calorimetric behaviors of the synthesized complexes, which are key properties of dispersed hydrocracking catalysts, were also explored. The experimental evaluation of the synthesized catalyst precursors show that the synthesized metal-based TBC[6] catalyst precursors improved the catalytic hydrogenation reactions. A co-catalytic system was also investigated by adding a commercial, first-stage hydrocracking supported catalyst in addition to the dispersed catalysts. The naphtha yields increased from 10.7 wt.% for the supported catalyst to 11.7 wt.% and 12 wt.% after adding it along with Ni-TBC[6] and Co-TBC[6], respectively. Mixing the metal-based precursors resulted in elevated yields of liquid products due to the in situ generation of highly active Co–Ni bimetallic dispersed catalysts.

**Keywords:** organometallic compounds; *p-tert*-butyl-calix[n]arene; dispersed catalysts; heavy oil upgrading; VGO

## 1. Introduction

Upgrading heavy petroleum feedstocks is essential to address the increased demand for light fuels in the coming years [1]. Hydrocracking is considered one of the trusted routes for upgrading heavy feedstocks into high-value light liquid products [2–4]. Catalytic hydrocracking can effectively remediate the detrimental challenge of condensation of the polynuclear aromatics during the cracking of large hydrocarbons by utilizing hydrogen during the cracking process [3,4]. Hydrocracking has been reported to achieve conversion levels of more than 95% for upgrading vacuum residue or heavy oil [5].

The slurry-phase hydrocracking process in particular is noteworthy for lowering coke formation, thus reducing the catalyst deactivation rate and the liquid product loss in the sediment [5]. An important advantage of the slurry process is the intimate contact between the feed and catalyst, which minimizes the mass transfer limitations. Moreover, it can deal with highly impure heavy feedstocks [6]. Moreover, the simplicity of the slurry-phase process allows it to handle highly contaminated feedstocks and yet at elevated efficiency [7]. The slurry-phase hydrocracking uses both unsupported soluble dispersed

catalysts and solid supported catalysts. However, the solid supported catalysts could experience significant catalyst deactivation due to coke deposition, resulting in equipment fouling and production losses [8]. A major advantage of the soluble dispersed catalysts is the flexibility in using them as standalone catalysts or in conjunction with supported catalysts. Soluble catalysts are classified into water-soluble and oil-soluble catalysts. Both classes have catalyst precursors that contain metals from groups IVB−VIIIB. Commonly used metals are Mo, Co, Cr, Ni, etc., typically in the form of salts or as ligands that are chemically linked to metals with organic compounds.

The soluble precursors are activated by reacting with $H_2S$, which results in the formation of the active phase of metal sulfide crystal [9,10]. The crystal particles achieve contact at the molecular level due to the small size of the crystals which guarantees high contact levels within the heavy oils, thus maximizing catalyst utilization [11]. The sulfidation can take place without pre-treatment as the catalyst is activated in situ upon contacting with the sulfur content already present in the oil feed after decomposing the organometallic structure at sufficiently high temperatures. The active metal sulfide crystals are formed from the reaction of the metallic cations released with the H−S obtained from the hydrodesulfurization of the oil feed containing organosulfur compounds. The sulfidation process takes place at 300−400 °C and a hydrogen pressure of 6.9−17.2 MPa, resulting in the formation of the infinitesimally small active metal sulfide crystals [8].

Oil-soluble organometallic precursors are the most prevalent candidates for dispersed catalysts due to the higher activity and dispersion capability with a higher surface-to-volume ratio in oil. This results in a boost in the catalytic hydrogenation reactions, which hinders the mesophase formation, an intermediate phase in cracking reactions, leading to aggregate formations and eventually coke deposition [12]. Oil-soluble precursors can be obtained from different organometallic compounds, such as organic acids (octoic, acetic, naphthenic, oxalic, etc.), metal salts of organic amines, and metal-containing quaternary ammonium compounds [11,12].

Although the active metal sulfide species from soluble precursors are formed in situ during the course of the reaction, the cost of precursor synthesis poses a challenge along with the difficulty of catalyst retrieval [4]. Iron-based catalyst counterparts, however, are the most widely used due to cost factors. Nevertheless, molybdenum-based catalysts generally show the most promising results due to the high activity of its sulfide form of $MoS_2$ in hydrogenation reactions in addition [8]. Another approach is to use bimetallic dispersed catalysts where cheaper promoters reduce the catalyst cost while contributing to hydrogenation reactions in a collaborative combination compared to the single metal ones [13–15].

Metallocalixarenes are considered to be good candidates for dispersed oil-soluble catalyst precursors, which have been proven to be successful for heavy oil upgrading [12,16]. Calixarenes are easily prepared with inexpensive materials, allowing them to be synthesized in sufficient quantities. Additionally, their favorable characteristics allow them to be environmentally friendly and attractive for catalytic applications in industrial processes and academic research activities (Table S1). Calixarenes are defined as oligomeric macrocyclic phenolic compounds that have been synthesized via the condensation of para-substituted phenols in the presence of formaldehyde at a specific temperature under alkaline conditions [17]. They can form complexes with neutral compounds as well as cations and anions. The calix[n]arenes can have different numbers of phenolic residues (n) involved in the structure, e.g., 4, 6, and 8. (Figure S1) [15]. They have generally high thermal stability ($T_m$ > 300 °C) and can sustain various chemical environments [18]. However, coordinating the host calixarene with metals decreases its thermal stability because of the decomposition of the complexes [19–21]. The limited degree of thermal stability for the metal-based calixarenes can positively be exploited in order to in situ generate dispersed hydrogeneration catalysts for processes operated at relatively severe conditions (e.g., ~400 °C).

As mentioned before, the calixarenes have been used previously with promising results. Thus, this work uses two in-house prepared oil-soluble dispersed catalyst precursors based

on a novel *p-tert*-butylcalix[6]arenes (TBC[6]) as a host for Ni and Co metals to (i) enhance the product yields of hydrocracking vacuum gas oil (VGO) and (ii) to study the promotional synergy effects of employing metal-based TBC[6] with a commercial first-stage bifunctional supported catalyst.

## 2. Results and Discussion

### 2.1. Characterization of Metal-Based p-tert-butylcalix[6]arenes

#### 2.1.1. Scanning Electron Microscopy-Energy Dispersive X-ray (SEM-EDX)

The SEM-EDX results of the synthesized Ni–TBC[6] catalyst are presented in Figure 1 using a magnification of 10,000×. The formed metal-based calixarene complex shows a surface with homogeneous crystal structure geometry, which is comparable to previous works (Figure 1a) [12,22]. Energy dispersive X-ray (EDX) of the sample reveals that Ni is being included by the calixarene host structure (Figure 1b). Additionally, the EDX elemental mapping shows that oxygen and carbon have a good homogeneity in the sample (Figure 1c). This observation was also similar to the Ni distribution in the sample.

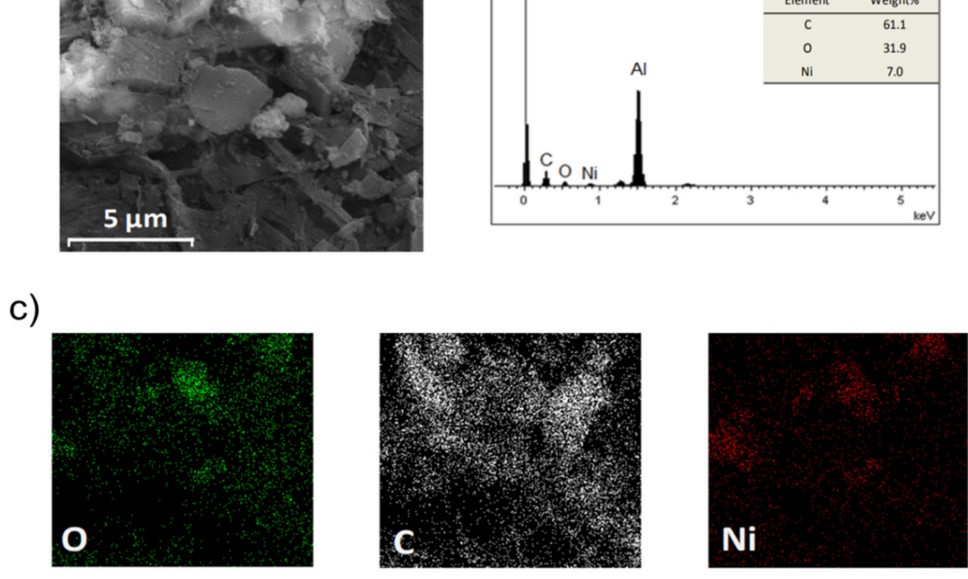

**Figure 1.** (**a**) SEM image, (**b**) EDX spectrum and (**c**) elemental mapping of Ni-TBC[6] at 10,000× magnification.

#### 2.1.2. Inductively Coupled Plasma (ICP)

The results of ICP for the Co-TBC[6] and Ni-TBC[6] reveal that the metal contents are 6 wt.% Co and 23 wt.% Ni, respectively. This result is in agreement with the EDX analysis discussed earlier. However, the ICP is considered to be more trusted since it is a bulk analysis, whereas EDX relies on the degree of homogeneity on the surface of the sample.

#### 2.1.3. Fourier Transform Infrared Spectroscopy (FT-IR)

Figure 2 reports the FT-IR spectra of parent TBC[6], Ni-TBC[6] and Co-TBC[6] samples. The free TBC[6] shows a characteristic vibrational band at 3132 cm$^{-1}$ related to the stretching of –OH groups of the cyclic hexamers. The phenomenon causing this stretching is the circular hydrogen bonding, which is due to the intramolecular hydrogen bonding. The vibrational band at 2966 cm$^{-1}$ is associated with the asymmetric –CH stretch, while the band at 2861 cm$^{-1}$ refers to its symmetric vibration. The characteristic absorption in the IR spectrum due to the aromatic stretching and vibration of –C=C– are identified with the peaks at 1602 cm$^{-1}$ and 1485 cm$^{-1}$, respectively. The bands in the fingerprint region at 1200 cm$^{-1}$ and 1155 cm$^{-1}$ and 1111 cm$^{-1}$ are in concordance with the stretching of the

C–O groups. The bands observed for the parent TBC[6] are in line with the spectra found in the literature [12,23].

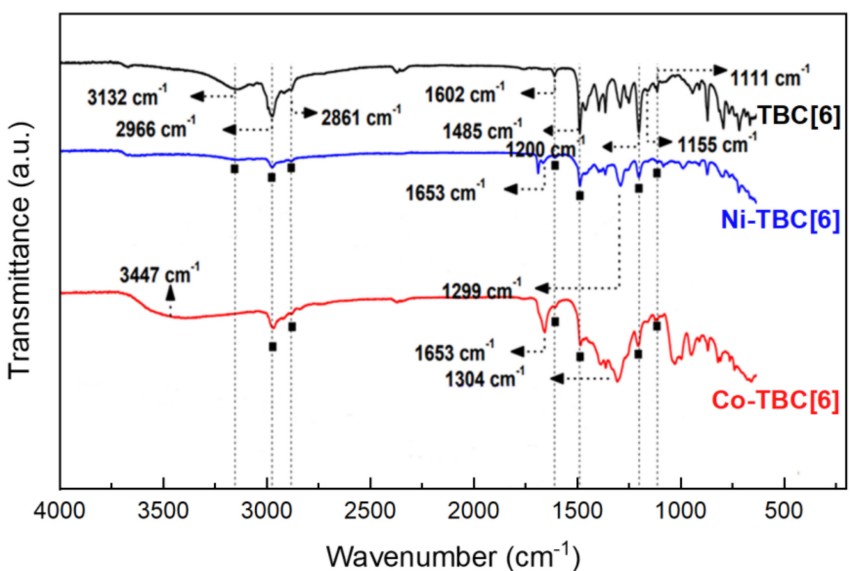

**Figure 2.** FT-IR spectra of TBC[6], Ni-TBC[6], and Co-TBC[6].

On the other hand, the spectra of the organometallic complexes show lower vibrational band intensities compared to their parent TBC[6]. The complexation with either $Ni^{2+}$ or $Co^{2+}$ lowered the intensity of –OH stretching of the host calixarene. The diagnostic region of the spectra (upper than 1500 cm$^{-1}$) of Ni-TBC[6], and Co-TBC[6] have the peaks at the same wavenumber in comparison with the parent molecule, in which case they have been identified using black squares. Furthermore, traces of DMF were identified with peaks around 1653 cm$^{-1}$ and 1300 cm$^{-1}$ [12,23].

2.1.4. Thermogravimetric/Calorimetric Analysis

The thermal decomposition study of the dispersed catalyst precursors is necessary to provide an insight into the mechanism of the metal release into the reaction medium and transformation to the active sulfide form. According to the literature [24], the parent calix[n]arene (n = 4, 6, and 8) decomposes at higher temperatures as the2 number of rings increases. Figure 3a,b show the thermal decomposition of synthesized metal-based catalyst precursors. The synthesized catalysts based on TBC[6] have shown different behaviors in their degradation. The case of Ni-TBC[6], and Co-TBC[6] showed decomposition in three stages with a total loss of 53 and 54 wt.%, respectively. The derivative thermogravimetric (DTG) profile shows that the highest speed of mass loss was reached at each stage at the different temperatures listed in Table S3. Overall, the results suggest that the synthesized metal-based TBC[6]s are more thermally stable than the four-membered metal-based calixarenes studied earlier [12].

As previous studies have discussed [12,24], the first peak that appears in the DTG between 57 and 280 °C is related to the release of $H_2O$, methanol, and DMF. Further increase in temperature leads to partial degradation of the organometallic structure with the second and third loss of mass. These mass losses are attributed to the release of $H_2O$, $CO_2$, and the *p-tert*-butyl group. Eventually, the oxidation of the organometallic compound will take place. This results in the formation of non-stable structures that ends up collapsing the organometallic structure. The heat flow profiles shown in Figure S2 provide information on the endothermic and exothermic processes during the structure annihilation of the metallocalixarenes. The fall in the heat flow before 100 °C is owing to reaching the glass transition temperature of calix[n]arenes molecules [25]. The endothermic peak of around a hundred degrees is related to the weight loss associated with the solvent release like

methanol and the humidity in the sample H$_2$O. Then, thermal decomposition continues to release DMF and that is also endothermic. After that, an exothermic peak appears due to the re-crystallization followed by another endothermic peak related to the melting process. After this point, the final degradation of the molecule happens. The more relative peaks of derivative heat flow described above are summarized in Table S4 for each catalyst and relate to the mass using DTG.

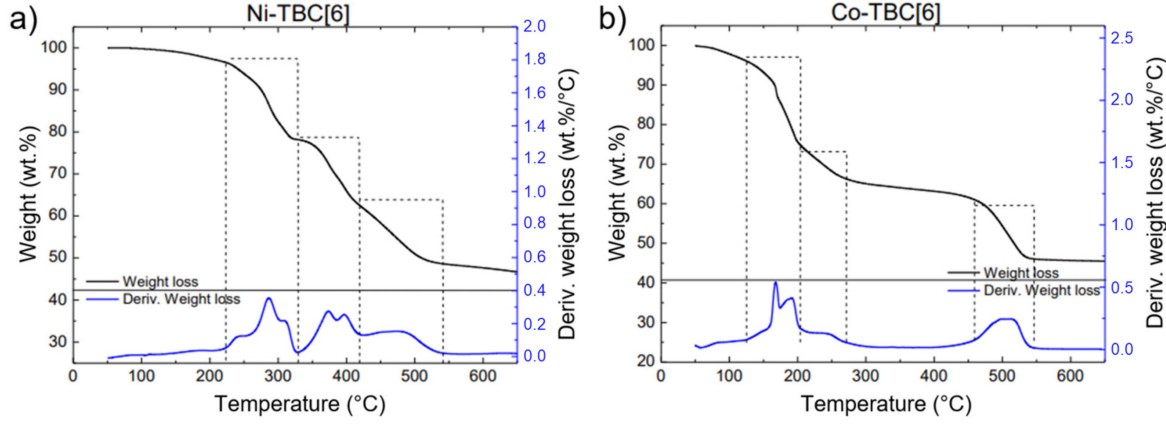

**Figure 3.** Thermogravimetric (TG) and differential thermogravimetric (DTG) profiles of (**a**) Ni-TBC[6] and (**b**) Co-TBC[6].

### 2.2. Performance Evaluation

The slurry-phase hydrocracking was conducted for the oil-soluble metal-based calixarene catalysts in a batch autoclave reactor. The catalyst precursors were evaluated for a standalone dispersed catalytic system and as an additive to a supported commercial hydrocracking catalyst.

### 2.2.1. Standalone Dispersed Catalysts

The performances of the synthesized Ni-TBC[6] and Co-TBC[6] complexes were tested as oil-soluble dispersed catalyst precursors with a 500 ppm concentration at a reaction temperature of 420 °C maintained for 1 h under a hydrogen pressure of 8.5 MPa. These conditions were found to be optimal to maintain the catalyst activity and minimize coke formation [26]. It is important to note that metal sulfides are active sites for hydrogenation/dehydrogenation reactions. The metal sulfidation takes place in situ due to the contact between the metal precursors released by the thermally annihilated metal-based calixarene structure and the sulfur-containing heteroatoms present in the vacuum gas oil (VGO) feedstock (Figure S3). Previous studies showed that the in situ formation of active metal crystals is independent of the organic ligand of the dispersed catalyst.

Figure 4 shows the product distribution classified into gases, naphtha, middle distillates, and vacuum gas oil (VGO) for the thermal run versus using the Ni-TBC[6] and Co-TBC[6]. The thermal hydrocracking showed a VGO conversion of 34.3%; however, it dropped for Ni-TBC[6] and Co-TBC[6] to 32.4% and 33%, respectively. This observation occurred because the dispersed catalysts are solely contributing to the hydrogenation reactions. The cracking reactions of the heavy hydrocarbon molecules predominantly take place due to thermal decomposition. The presence of the metal sulfides promotes the hydrogenation/dehydrogenation reactions. Therefore, the addition of the dispersed catalysts increases the liquid product fractions and decreases the coke formation. The use of Ni-TBC[6] and Co-TBC[6] as standalone dispersed catalysts produced total liquid yields of 28.4 wt.% and 28.8 wt.%, which are comparable to that reported for the thermal run (29.9 wt.%). Nevertheless, mixing the Ni-TBC[6] and Co-TBC[6] were found to have the highest naphtha and distillate yields compared to the sole dispersed catalysts with a total

liquid yields of ~32 wt.%. This is due to the formation of a bimetallic active phase such as Co–Ni–S that offers higher sulfur conversions and more liquid fractions [27].

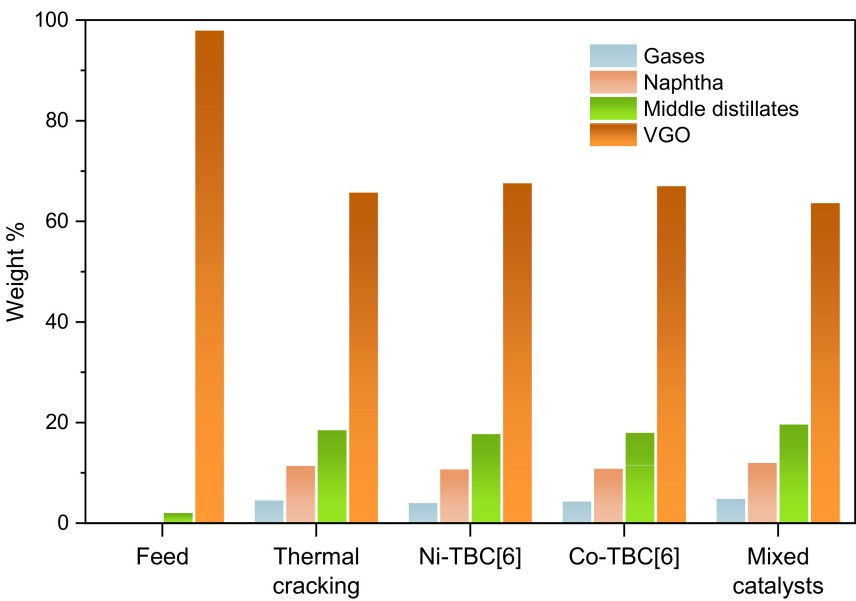

**Figure 4.** Product yield distribution of VGO hydrocracking at 420 °C under an $H_2$ pressure of 8.5 MPa for a reaction time of 1 h for dispersed metal catalysts. The dispersed catalysts' metal concentration for Ni-TBC[6], Co-TBC[6], and mixed catalysts were 500 ppm for each case. Coke yield for all cases was negligible.

### 2.2.2. Dispersed Catalysts with Supported Catalyst

A commercial first-stage hydrocracking catalyst ($W-Ni/Al_2O_3-SiO_2$) was used for the co-catalytic experiments to demonstrate the promotional effects of the dispersed catalysts. The properties of the supported commercial $W-Ni/Al_2O_3-SiO_2$ catalyst are listed in Table S5. The supported catalyst-to-oil ratio was (1:20) with the dispersed catalyst precursor concentration of 500 ppm. The product distribution for the experimental runs is shown in Figure 5. Introducing 5 wt.% of the supported has generally increased the VGO conversion compared to the results obtained with the standalone dispersed catalysts. This is due to the presence of acidic sites on the supported catalyst, which promotes further catalytic cracking.

The VGO conversion for the commercial-grade supported catalyst alone was 39%, which further improved when Ni-TBC[6] and Co-TBC[6] are introduced to 41.3% and 42.4%, respectively. The naphtha yields increased from 10.7 wt.% for the supported catalyst to 11.7 wt.%, 12 wt.%, and 12.1 wt.% for Ni-TBC[6] and Co-TBC[6], respectively. The cracked intermediates were hydrogenated on the active sites of both the supported and dispersed catalysts. Further cracking followed by hydrogenation of the cracked intermediates is reflected by increased amounts of the liquid products.

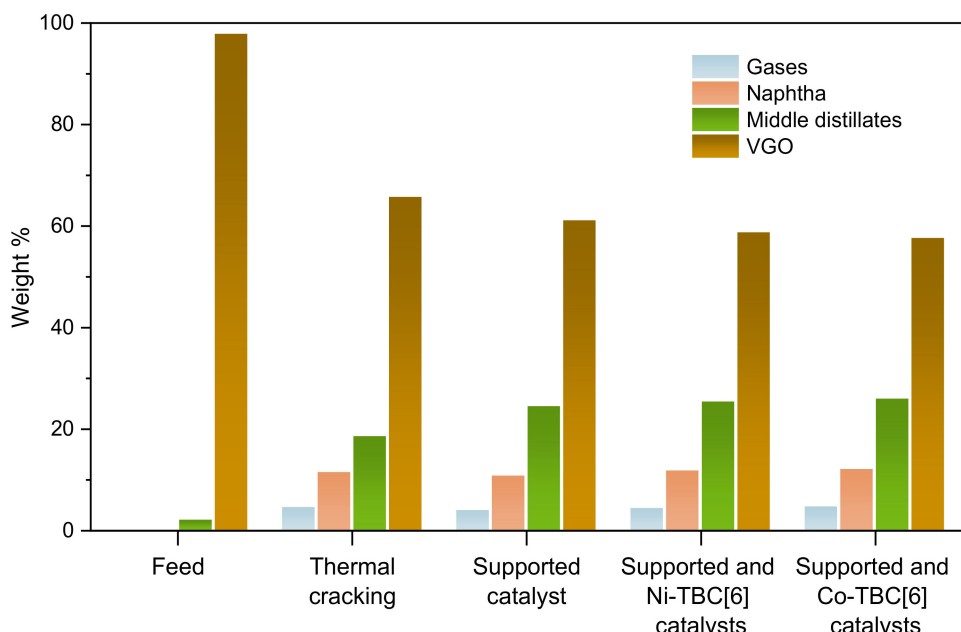

**Figure 5.** Product yield distribution and conversion of VGO hydrocracking at 420 °C under an $H_2$ pressure of 8.5 MPa for a reaction time of 1 h for supported and dispersed metal catalysts. The dispersed catalysts metal concentrations for Ni-TBC[6] and Co-TBC[6] catalysts were 500 ppm for each case. Coke yield for all cases was negligible.

## 3. Materials and Methods

### 3.1. Materials

In this work, nickel(II) nitrate hexahydrate ($Ni(NO_3)_2 \cdot 6H_2O$, 99.999% trace metals basis), cobalt(II) nitrate hexahydrate ($Co(NO_3)_2 \cdot 6H_2O$, 4-*tert*-butylcalix[6]arene ($C_{44}H_{56}O_4$, ≥99.0%), reagent grade, ≥98.0%), N,N-dimethylformamide ($HCON(CH_3)_2$, anhydrous, 99.8%), triethylamine (($C_2H_5)_3N$, ≥99.5%), dimethyl sulfoxide (($CH_3)_2SO$, reagent grade, 99.5%), and methanol ($CH_3OH$ anhydrous, 99.8%) were used to synthesize the metal-based calixarenes. All chemicals were obtained from Sigma-Aldrich, USA, and used without further purification. The commercial hydrocracking catalyst used was $W-Ni/Al_2O_3-SiO_2$ and its properties are shown in Table S5.

The feedstock for the hydrocracking process (vacuum gas oil; VGO) was received from a Saudi Aramco Refinery. Table S2 shows the chemical and physical properties of the VGO. Nitrogen and hydrogen gases used during the catalyst evaluation were purchased from a local supplier with 99.999% purity.

### 3.2. Synthesis of Metal-Based p-tert-butylcalix[6]arenes

The metallocalixarenes were prepared from the parent *p-tert*-butylcalix[6]arene, hereafter referred to as TBC[6]. The precursor of the Co and Ni metals were cobalt(II) nitrate hexahydrate and nickel(II) nitrate hexahydrate, respectively. The calixarene structure was synthesized by adding 100 mg of TBC[6] to 10 mL dimethylformamide (DMF) followed by heating and stirring at 60 °C until forming a colloidal solution. Then, 0.6 mL of triethylamine was added until the mixture turned transparent. Two grams of the metal precursor were added to 10 mL of methanol followed by drop-wise addition of 1.5 mL dimethyl sulfoxide (DMSO) while stirring to make the metal precursor solution. The solutions were mixed and stirred in an ice bath at 4 °C for 24 h. The prepared metal-based calixarenes were obtained by filtration using a Millipore nylon membrane (Isopore Membrane Filters) with a pore size of 0.6 μm.

### 3.3. Catalyst Characterization

To determine the metal content in the metal-based *p-tert*-butylcalix[6]arene complexes accurately, inductivity coupled plasma (ICP) was carried out in a PlasmaQuant PQ 9000 (Analytik Jena GmbH, Jena, Germany). The catalysts were digested in 65% $HNO_3$. Ten milligrams of each sample were mixed with 5.0 mL of $HNO_3$ at $60-70\ ^\circ$C until the total volume was reduced to 2.0 mL through evaporation of excess $HNO_3$. Then, the digested solution was cooled down and its volume rose to 30 mL by adding deionized water. The solution was heated to $50\ ^\circ$C for 1.5 h. The solution was filtered using a 0.1 μm filter paper (MilliporeSigma, Burlington, MA, USA). The filtrate volume was increased to 50.0 mL by adding deionized water.

The catalyst morphology was analyzed by scanning electron microscopy (SEM) to obtain a visual idea of the physical surface of the catalyst. The analyses were conducted using a JSM-6460LV (JEOL Ltd., Akishima, Tokyo, Japan) scanning electron microscope operated at an acceleration voltage of 20 kV. This system was combined with energy dispersive X-ray (EDX) spectroscopy to identify the elemental composition of the samples. Each sample was coated with gold (5 nm thickness) on a sputter coating machine before being placed in the holder to undergo bombardment by electrons. The electron beam is accelerated by high voltage (20 kV in this case) and goes through a system of openings and electromagnetic lenses until a very fine electron beam is 4 produced. This beam scans the material surface, which is observed on the microscope screen, obtaining the final image.

Fourier transform infrared (FT-IR) spectroscopy was conducted to elucidate and confirm the molecular structure of the metal-based calixarenes. The equipment used was Nicolet iS5, Thermo Scientific, Waltham, MA, USA.

Finally, the thermal stability of the catalysts has been studied by thermogravimetric and calorimetric studies were performed using an SDT Q600 simultaneous DSC/TGA analyzer (TA Instruments, New Castle, DE, USA). The thermograms were examined to study the heat flow changes, either exothermic or endothermic, in the samples with temperature relative to that of an inert reference, that is, sapphire calibrant. The analysis was performed from room temperature to $600\ ^\circ$C at a $4\ ^\circ$C/min heating rate under 100 mL/min nitrogen flow.

### 3.4. Hydrocracking Experimental Conditions

The reactor used for testing the catalytic performance of the prepared complexes was a 300 mL batch autoclave reactor (Parker Autoclave Engineers, Erie, PA, USA) (Figure 6). Each experiment starts with feeding the vessel of batch reactor with the desired amount of the VGO and catalyst precursor at room temperature. Then, after pass the leak test, the reactor is gradually heated to desired reaction temperature using a heating jacket. For this step, reactor has been fed with hydrogen at 3 MPa to minimize the probability of the reactions taking place during the heating period. After that, the hydrogen pressure is then increased to 8.5 MPa for the hydrocracking reaction and maintained throughout the course of reaction. Additionally, the agitator is started at 950 rpm to ensure the homogeneity of the feed with the hydrogen and the catalyst. The amount of gas produced during the reactor is calculated from the difference in weight of the liquid product and the fresh feed.

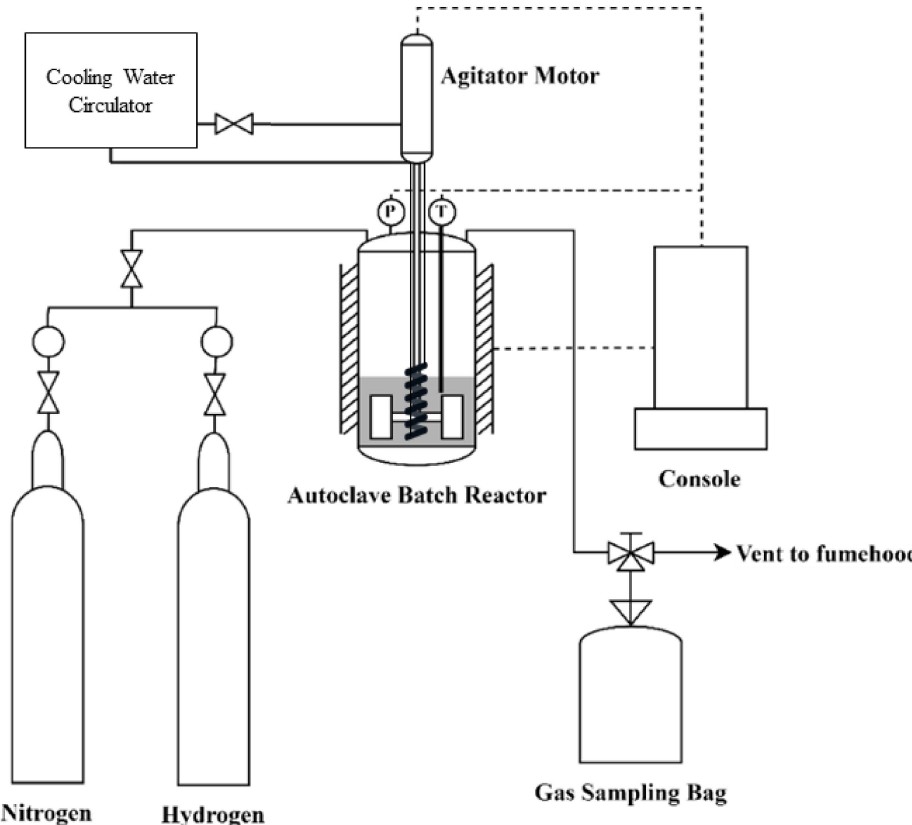

**Figure 6.** Schematic of the autoclave batch reactor used for the hydrocracking of VGO experiments.

Products Analysis

The thermogravimetric analyzer (TGA) was used in this study to evaluate the product distribution of the hydrocracking. The products of the hydrocracking experiment were divided into lumps based on their boiling point ranges as used by the previous studies reported in the literature [26,28–32]. Five lumps were considered based on distillation boiling ranges: gases, <90 °C; naphtha, 90–221 °C; middles distillate, 221–343 °C; and VGO, 343–565 °C. For each sample analyzed, ~70 mg of the liquid product is put on the microbalance of the TGA in the temperature range of 50–600 °C under nitrogen flowing at 100 mL/min with a 10 °C/min heating rate.

The conversion of VGO is calculated by Equation (1) as follows:

$$\text{conversion (wt.\%)} = \frac{W_{VGO_0} - W_{VGO}}{W_{VGO_0}} \times 100 \tag{1}$$

where $W_{VGO_0}$ and $W_{VGO}$ are the weight of VGO fed initially and remained after the process, respectively. The yield of a product is defined based on its weight percentage from the total effluent as shown in Equation (2):

$$Y_i \text{ (wt.\%)} = \frac{W_i}{W_p} \times 100 \tag{2}$$

where $W_i$ is the weight of the product (i.e., gases, naphtha, distillate, VGO, or coke) and $W_p$ is the weight of the total product [33].

## 4. Conclusions

Following are the conclusions of this experimental investigation:

I.　The synthesis of the metal-based TBC[6] dispersed catalysts was achieved successfully and characterized by different techniques including ICP, FT-IR, SEM, and TG-

DSC. The results confirm the cation coordination with the TBC[6] ligand to form organometallic compounds.

II. The DSC profile of the TBC[6] shows three thermal decomposition stages, which can be exploited to release the metal precursors and subsequently generate the active dispersed phase.

III. The VGO hydrocracking tests show that the use of mixed Ni- and Co-TBC[6] dispersed catalyst precursors is yielding the highest naphtha and distillate fractions (12 wt.% and 19.6 wt.%) due to the in situ generation of highly active Co–Ni bimetallic dispersed catalyst.

IV. The dispersed catalyst precursors offer easy access to the hydrogenation sites for the intermediate hydrocarbon molecules and reactive hydrogen species, which enhances the hydrogenation reactions and lowers coke formation and catalyst deactivation.

V. The use of dispersed and supported co-catalytic configuration increased naphtha yields from 10.7 wt.% for the supported catalyst to 11.7 wt.%, 12 wt.%, and 12.1 wt.% for Ni-TBC[6], Co-TBC[6] and Ni-Co-TBC[6], respectively. This observation can be explained due to the contribution of the supported catalyst acidic sites, which participate in the hydrogenation of the cracked molecules.

**Supplementary Materials:** The following supporting information can be downloaded at: https://www.mdpi.com/article/10.3390/catal12101255/s1, Figure S1: Structures of p-tert-butylcalix[n]arenes [24].; Figure S2: Differential thermogravimetric (DTG) and Differential heat flow profiles of (a) Ni-TBC[6] and (b) Co-TBC[6]; Figure S3: Mechanism of in situ sulfidation to form the active metal sulfide from metal-based calixarenes; Table S1: Features and Advantages Offered by Calixarene [11]; Table S2: Physical and chemical properties of the vacuum gas oil (VGO); Table S3: Temperature of peaks in DTG; Table S4: Characteristic peaks in DSC; Table S5: Properties of the Commercial Hydrocracking Catalyst (KC-2710).

**Author Contributions:** Conceptualization, M.M.H., H.Z., M.I., F.A.A.-Z. and F.J.D.; formal analysis, M.I., F.A.A.-Z., F.J.D. and M.A.B.S.; investigation, M.I., F.A.A.-Z., H.Z. and F.J.D.; writing—original draft preparation, M.I. and F.J.D.; writing—review and editing, M.M.H. and T.A.-A.; supervision, M.M.H.; project administration, S.A.A., M.A.B.S. and M.M.H.; funding acquisition, M.M.H. All authors have read and agreed to the published version of the manuscript.

**Funding:** This research was funded by Deanship of Research Oversight and Coordination (DROC) at King Fahd University of Petroleum and Minerals (KFUPM), grant number DF181018.

**Data Availability Statement:** The data presented in this study are available in this paper.

**Acknowledgments:** The author(s) would like to acknowledge the support provided by the Deanship of Research Oversight and Coordination (DROC) at King Fahd University of Petroleum and Minerals (KFUPM) for funding this work through project. No. DF181018.

**Conflicts of Interest:** The authors declare no conflict of interest.

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
