# Peer review of "Experimental Investigation of Metal-Based Calixarenes as Dispersed Catalyst Precursors for Heavy Oil Hydrocracking"

_catalysts, doi:10.3390/catal12101255_

Round 1

Reviewer 1 Report

Dear Editor,

I'm very glad to be the review of this paper (Manuscript Number ID: catalysts-1948457). Title: Experimental investigation of metal-based calixarenes as dispersed catalyst precursors for heavy oil hydrocracking. I decided to accept this paper for the publishing in Journal of Catalysts (ISSN 2073-4344) after major correction with following comments:

1. I could not found a recognizing contribution in this paper there are plenty of results, but the authors were not focusing on the novelty of the paper. 

2. I have not found a comparative study with other catalyst such as SBA-15, MCM-41, MCM-48, and so on. Drug Delivery, 28 (1) (2021) 856–864. https://doi.org/10.1080/10717544.2021.1914778.

3.  English should be improved throughout the manuscript. 

4.  The abstract must be rewritten again with a reduction and explaining the major finding with a conclusion. Journal of Molecular Structure 1260 (2022) 132879.

5. Re-write the introduction part to give more details. 

6.  The introduction part must be containing modern references such as: 

Microporous and Mesoporous Materials 341 (2022) 112020. https://doi.org/10.1016/j.micromeso.2022.112020.

7.  The aims of the present work must be more cleared at the end of the introduction part.

9. The surface area is very important for catalyst. The author must be discussing this point in detail because the surface area is a very important factor in increasing the reaction rate.

10. The author must be clear about the experimental steps for the preparation of calibration curves and stock solutions.

11. I hope the author draws the schematic diagram for all the processes.

12. Is the Hydrophilic or Hydrophobic catalyst performed in this study and Why?

Catalysts 2022, 12(3), 324. https://doi.org/10.3390/ catal12030324.

13. What is the relationship between surface roughness and contact angle for the Co- and Ni-based TBC catalyst. The author must be discussing this point in detail.

14. The author must be clear about the experimental steps and discuss the interfacial phenomena for this study.

15. What is the main conclusion of this study? The conclusion must be reduced. 

16. Maybe the leaching occurred from the surface of the catalyst Co- and Ni-based TBC materials that mean dissolved in the solution. I need the author to explain this important point in the result and discussion part. 

17. The Co- and Ni-based TBCcatalyst material characterizations must appear according to the sequence XRD, BET surface area, SEM, FT-IR, and TGA. Heliyon 8 (2022) e09737. https://doi.org/10.1016/j.heliyon.2022.e09737.

18. Why the author used a Co- and Ni-based TBC instead of other catalyst.

19. I hope the author of this manuscript achieves a comparison between this study and others the same reaction.

22. What is the purpose from Kinetics study such as; pseudo-first, second-order, and Intraparticle diffusion was included in this study.

Author Response

Reviewer - 1

Comments and Suggestions for Authors

Dear Editor,

I'm very glad to be the review of this paper (Manuscript Number ID: catalysts-1948457). Title: Experimental investigation of metal-based calixarenes as dispersed catalyst precursors for heavy oil hydrocracking. I decided to accept this paper for the publishing in Journal of Catalysts (ISSN 2073-4344) after major correction with following comments:

  • Response:

We would like to thank the Reviewer for his/her comment and recommendation to accept our work in Catalysts after major correction.

  1. I could not found a recognizing contribution in this paper there are plenty of results, but the authors were not focusing on the novelty of the paper.
  • Response:

The authors appreciate the comment of the Reviewer. We have modified the last paragraph of the introduction to clarify the novelty of this work. The last paragraph of the introduction (line 96-104) has been modifying as follow:

“… The limited degree of thermal stability for the metal-based calixarenes can positively be exploited in order to in situ generate dispersed hydrogeneration catalysts for processes operated at relatively severe conditions (e.g. ~ 400°C).

As mentioned before, the calixarenes have been used previously with promising results. Thus, this work uses two in-house prepared oil-soluble dispersed catalyst pre-cursors based on a novel p-tert-butylcalix[6]arenes (TBC[6]) as a host for Ni and Co metals to (i) enhance the product yields of hydrocracking vacuum gas oil (VGO) and (ii) to study the promotional synergy effects of employing metal-based TBC[6] with a commercial first-stage bifunctional supported catalyst.”

  1. I have not found a comparative study with other catalyst such as SBA-15, MCM-41, MCM-48, and so on. Drug Delivery, 28 (1) (2021) 856–864. https://doi.org/10.1080/10717544.2021.1914778.
  • Response:

Authors understand the importance of compare catalysts as reviewer point out and we have done with different configurations of homogeneous catalyst, which is the subject of this work. Nevertheless, the mentioned examples by the Reviewer are falling under a different category of catalysts, which is solid supported catalysts (heterogeneous catalysis).

  1. English should be improved throughout the manuscript.
  • Response:

We have revised in depth the manuscript to improve the English throughout them.

  1. The abstract must be rewritten again with a reduction and explaining the major finding with a conclusion. Journal of Molecular Structure 1260 (2022) 132879.
  • Response:

The abstract has been revised and adapted as proposed by the Reviewer.

  1. Re-write the introduction part to give more details.
  • Response:

The introduction part has been modified to give more details in the revised version of the manuscript.

  1. The introduction part must be containing modern references such as:

Microporous and Mesoporous Materials 341 (2022) 112020. https://doi.org/10.1016/j.micromeso.2022.112020.

  • Response:

The following references have been added throughout the manuscript:

  • Fuel 321 (2022) 124029
  • Catalysts (2022), 12, 1125
  • Fuel 315 (2022) 123134
  • Fuel 288 (2021) 119686

  1. The aims of the present work must be more cleared at the end of the introduction part.
  • Response:

We have modified the last paragraph of the introduction to be as follow:

“…As mentioned before, the calixarenes have been used previously with promising results. Thus, this work uses two in-house prepared oil-soluble dispersed catalyst pre-cursors based on a novel p-tert-butylcalix[6]arenes (TBC[6]) as a host for Ni and Co metals to (i) enhance the product yields of hydrocracking vacuum gas oil (VGO) and (ii) to study the promotional synergy effects of employing metal-based TBC[6] with a commercial first-stage bifunctional supported catalyst.”

  1. The surface area is very important for catalyst. The author must be discussing this point in detail because the surface area is a very important factor in increasing the reaction rate.
  • Response:

Authors agree with the reviewer but in heterogeneous catalysis. However, here metal-based calixarenes are used in this study as homogeneously dispersed catalyst precursors to in situ form metal sulfide active hydrogenation sites after thermal annihilation. Therefore, surface area is not important on these catalysts. Please refer to the answer of comment# 11 and 16. Therefore, the surface area of the organometallic structure is irrelevant in this matter.

  1. The author must be clear about the experimental steps for the preparation of calibration curves and stock solutions.
  • Response:

The Reviewer’s question is not specific and clear. Nevertheless, if we assume calibration curves are related to the TGA/DSC, we have added the following elaboration under section 2.3 (line 158-160) of the revised manuscript:

“…The thermograms were examined to study the heat flow changes, either exothermic or endothermic, in the samples with temperature relative to that of an inert reference, that is, sapphire calibrant.”

  1. I hope the author draws the schematic diagram for all the processes.
  • Response:

According to the reviewer’s request, the following scheme has been added in the revised Supplementary Materials to clarify the in situ sulfidation process to generate the hydrogenation active sites. The figure has been cited in the revised version of the manuscript under section 3.2.1:

Figure S3. Mechanism of in situ sulfidation to form the active metal sulfide from metal-based calixarenes.

  1. Is the Hydrophilic or Hydrophobic catalyst performed in this study and Why?

Catalysts 2022, 12(3), 324. https://doi.org/10.3390/ catal12030324.

  • Response:

The Reviewer’s question is not clear. However, if we assume the question is regarding performing contact angle experiment, we have not done since we believe it is irrelevant to this work.

  1. What is the relationship between surface roughness and contact angle for the Co- and Ni-based TBC catalyst. The author must be discussing this point in detail.
  • Response:

Please refer to the previous answer of question number 12.

  1. The author must be clear about the experimental steps and discuss the interfacial phenomena for this study.
  • Response:

Following the suggestion of the reviewer, authors clarify the experimental steps to enhance the comprehension of the reaction system and procedure. Regarding the interfacial phenomena, the respected Reviewer needs to be more specific on his/her request since it is not clear. Authors modify the first paragraph of Section 2.4 as follow:

“…was a 300 mL batch autoclave reactor (Parker Autoclave Engineers, USA) (Figure 1). Each experiment starts with feeding the vessel of batch reactor with the desired amount of the VGO and catalyst precursor at room temperature. Then, after pass the leak test, the reactor is gradually heated to desired reaction temperature using a heating jacket. For this step, reactor has been fed with hydrogen at 3 MPa to minimize the probability of the…”

  1. What is the main conclusion of this study? The conclusion must be reduced.
  • Response:

The main conclusion of the study is that metal-based calixarene can be used as a standalone dispersed catalyst precursors or a co-catalyst besides supported bifunctional catalysts to enhancing catalytic hydrogenation reactions. Moreover, mixing the metal-based precursors resulted in elevated yields toward liquid products due to the in situ generation of highly active Co–Ni bimetallic dispersed catalyst. The conclusion has been reduced in the revised manuscript.

  1. Maybe the leaching occurred from the surface of the catalyst Co- and Ni-based TBC materials that mean dissolved in the solution. I need the author to explain this important point in the result and discussion part.
  • Response:

The discussion on the in situ sulfidation reaction has been modified and elaborated under section 3.2.1 in the revised manuscript to be as follows:

“…It is important to note that metal sulfides are active sites for hydrogenation/ dehydrogenation reactions. The metal sulfidation takes place in situ due to the contact between the metal precursors released by the thermally annihilated metal-based calixarene structure and the sulfur-containing heteroatoms present in the vacuum gas oil (VGO) feedstock (Figure S3). Previous studies showed that the in situ formation of active metal crystals is independent of the organic ligand of the dispersed catalyst.”

  1. The Co- and Ni-based TBCcatalyst material characterizations must appear according to the sequence XRD, BET surface area, SEM, FT-IR, and TGA. Heliyon 8 (2022) e09737. https://doi.org/10.1016/j.heliyon.2022.e09737.
  • Response:

We respect the Reviewer’s viewpoint. However, we believe the current sequence of the material characterization techniques is adequate.

  1. Why the author used a Co- and Ni-based TBC instead of other catalyst.
  • Response:

We have added the following statement in the revised version of the manuscript to elaborate on the motivation of using metal-based TBCs as dispersed catalyst precursors in this study:

“Calixarenes are easily prepared with inexpensive materials, allowing them to be synthesized in sufficient quantities. Also, their favorable characteristics allow them to be environment-friendly and attractive for catalytic applications in industrial processes and academic research activities (Table S1).”

  1. I hope the author of this manuscript achieves a comparison between this study and others the same reaction.
  • Response:

To be able to compare our catalytic performance with literature, we need to estimate kinetic parameters of the process (as explained in this next answer of comment #22). Please refer to our previously published article for more details:

  • Energy Fuels )2017(, 31, 3132−3142
  • Fuel Processing Technology 185 (2019) 158–168

Therefore, we would like to mention that the kinetic modelling is planned to be conducted and published in a separate work.

  1. What is the purpose from Kinetics study such as; pseudo-first, second-order, and Intraparticle diffusion was included in this study.
  • Response:

We have not included kinetic study in this manuscript. However, kinetic modeling is a major tool to further commenting on the enhancement of the catalytic hydrogenation/dehydrogenation reactions governed upon introducing the dispersed catalyst. Moreover, it can be useful to investigate the synergy between the solid supported catalysts and dispersed.

Reviewer 2 Report

Dear Editor

The article entitled “Experimental investigation of metal-based calixarenes as dispersed catalyst precursors for heavy oil hydrocracking” has been reviewed. In this manuscript, the authors employed metal-containing oil-soluble compounds as precursors of catalysts for heavy oil upgrading via hydrocracking. They synthesized metal-based p-tert-butylcalix[6]arene (TBC[6]s) organic species and used as dispersed catalyst precursors to enhance catalytic hydrogenation reactions. The subject has industrial importance and can be considered for publication after a minor revision as below:

1-   It is suggested the authors declare clearly the novelty of research at the end of Introdcution.

2-   In Figure 1, the inlet for raw material entrance did not included in the Scheme. How they feed catalyst and oil inside the reactor?

3-   Figure 2, It seems that the synthesized catalyst does not show desired spherical morphology. Actually it does not have any specific shape. It is recommended the authors clarify this issue. Also, in the EDX spectrum, the amount of Al was not specified.

4-   Do the authors have any suggestion about the reusability of the catalyst?

Author Response

Reviewer – 2

Comments and Suggestions for Authors

Dear Editor

The article entitled “Experimental investigation of metal-based calixarenes as dispersed catalyst precursors for heavy oil hydrocracking” has been reviewed. In this manuscript, the authors employed metal-containing oil-soluble compounds as precursors of catalysts for heavy oil upgrading via hydrocracking. They synthesized metal-based p-tert-butylcalix[6]arene (TBC[6]s) organic species and used as dispersed catalyst precursors to enhance catalytic hydrogenation reactions. The subject has industrial importance and can be considered for publication after a minor revision as below:

  • Response:

We would like to thank the Reviewer for his/her comment and recommendation to publish our work in Catalysts.

1-   It is suggested the authors declare clearly the novelty of research at the end of Introdcution.

  • Response:

The authors appreciate the comment of the Reviewer. We have modified the last paragraph of the introduction to clarify the novelty of this work. The last paragraph of the introduction (line 96-104) has been modifying as follow:

“… The limited degree of thermal stability for the metal-based calixarenes can positively be exploited in order to in situ generate dispersed hydrogeneration catalysts for pro-cesses operated at relatively severe conditions (e.g. ~ 400°C).

As mentioned before, the calixarenes have been used previously with promising results. Thus, this work uses two in-house prepared oil-soluble dispersed catalyst pre-cursors based on a novel p-tert-butylcalix[6]arenes (TBC[6]) as a host for Ni and Co metals to (i) enhance the product yields of hydrocracking vacuum gas oil (VGO) and (ii) to study the promotional synergy effects of employing metal-based TBC[6] with a commercial first-stage bifunctional supported catalyst.”

2-   In Figure 1, the inlet for raw material entrance did not included in the Scheme. How they feed catalyst and oil inside the reactor?

  • Response:

As is mentioned in Section 2.4, it is a batch reaction system. Therefore, both VGO and catalyst were fed from the top of the vessel of the reactor then the reactor is fitted, and the process was run in batch mode. The following statement has been added in line 166 to clarify:

“…with feeding the vessel of batch reactor with the desired amount of the VGO and cata-lyst precursor.”

3-   Figure 2, It seems that the synthesized catalyst does not show desired spherical morphology. Actually it does not have any specific shape. It is recommended the authors clarify this issue. Also, in the EDX spectrum, the amount of Al was not specified.

  • Response:

The observation made by the Reviewer is rightly done and the authors agree with it. Emphasis has been placed in the manuscript about the morphology obtained by SEM is homogeneous crystal structures to clarify this aspect. This phrase is the one that follows (line 199):

“...in Figure 2 using a magnification of ×10000. The formed metal-based calixarene complex shows a surface with homogeneous crystal structure geometry, which is comparable to previous works (Figure 2a) [10,25]. Energy dispersive X-ray (EDX) of the…”

On the other hand, Figure 2B has been modified as requested by the Reviewer to be as follows:

Al was not included in the calculation, as it was not part of the catalyst – it appeared from the sample holder aluminum foil.

4-   Do the authors have any suggestion about the reusability of the catalyst?

  • Response:

The authors appreciate the suggestion raised by the Reviewer. Nevertheless, oil-soluble catalysts cannot be recovered after the reaction since they remain dissolved in the liquid products.

Reviewer 3 Report

It is a nice and interesting paper on dispersed catalyst precursors for heavy oil hydrocracking.

The paper merits publication once the authors have correctly addressed the following questions:

·         In the Hydrocracking experimental conditions section, the authors say that “The hydrogen pressure is then increased to 8.5 MPa for the hydrocracking reaction”. It is not clear if the hydrogen is fed continuously and regulated to maintain that pressure or if it is a batch reactor and the initial pressure of 8.5 MPa varies with reaction time. They should clear it up.

·         In Figure 3, it is not known which is the spectrum of Ni-TBC[6] and Co-TBC[6] since the legend in two of the spectra is the same (Ni-TBC[6]).

·         The catalysts have a metal content of 6 wt% Co and 23 wt% Ni. What is the difference? What is the stoichiometric amount of Co and/or Ni that calixarenes can contain?

·         In Table S3, endothermic peaks 2 and 3 in the case of Ni-TBC[6] are not really seen in Figure S2. It would be better to remove them from the table. There is a typo in the temperature of peak 6 corresponding to Co-TBC[6]: it is not 5230 ºC but 523 ºC.

·         The authors should have connected the exhaust gas line of the thermogravimetric equipment on-line with a mass spectrometer to try to find out which compounds are released during the TGA.

·         In section 3.2.1 it is said that the addition of the dispersed catalysts increases the liquid product fractions and decreases the coke formation. However, two lines below it is shown that the yield of liquids with these catalysts is lower than that obtained by thermal cracking.

·         Figures 5 and 6 should also show the yield to coke or at least say that it is negligible.

·         In section 3.2.2 it is said that the properties of the supported commercial catalyst are shown in Table S4. However in that Table S4, the properties of the VGO are shown. That is, in the supplementary material the properties of that commercial catalyst are not shown.

·         Perhaps, the most interesting thing would have been to analyze the composition of the liquid fractions from thermal hydrocracking and catalytic hydrocracking to compare their composition and really see the usefulness of using these catalysts.

Author Response

Reviewer – 3

Comments and Suggestions for Authors

It is a nice and interesting paper on dispersed catalyst precursors for heavy oil hydrocracking. The paper merits publication once the authors have correctly addressed the following questions:

  • Response:

We would like to thank the Reviewer for his/her comment and recommendation to publish our work in Catalysts.

  • In the Hydrocracking experimental conditions section, the authors say that “The hydrogen pressure is then increased to 8.5 MPa for the hydrocracking reaction”. It is not clear if the hydrogen is fed continuously and regulated to maintain that pressure or if it is a batch reactor and the initial pressure of 8.5 MPa varies with reaction time. They should clear it up.
  • Response:

Hydrogen is fed firstly at lower pressure during heating up the reactor to minimize the probability of the reactions taking place during the heating period. Then, the pressure is increased once the temperature of reaction is reached and maintained throughout the course of reaction. The following statement has been added in line 170 of the revised version of the manuscript:

“The hydrogen pressure is then increased to 8.5 MPa for the hydrocracking reaction and maintained throughout the course of reaction.”

  • In Figure 3, it is not known which is the spectrum of Ni-TBC[6] and Co-TBC[6] since the legend in two of the spectra is the same (Ni-TBC[6]).
  • Response:

We thank the Reviewer on this comment. Figure 2B has been modified as requested by the Reviewer to be as follows:

  • The catalysts have a metal content of 6 wt% Co and 23 wt% Ni. What is the difference? What is the stoichiometric amount of Co and/or Ni that calixarenes can contain?
  • Response:

The presence of phenoxy groups in calix[6]arenes enables transition-metal cations to form metal phenolate complexes by substituting for one to six hydrogen atoms. However, we believe the difference in metal content for the formed complexes is due to decreasing cation size, that is, Co2+ > Ni2+.

  • In Table S3, endothermic peaks 2 and 3 in the case of Ni-TBC[6] are not really seen in Figure S2. It would be better to remove them from the table. There is a typo in the temperature of peak 6 corresponding to Co-TBC[6]: it is not 5230 ºC but 523 ºC.
  • Response:

We thank the Reviewer for the comments on Table S3. We fixed the table as suggested as follows:

Peak (°C)

Ni-TBC[6]

Co-TBC[6]

1

75a

-

2

-

168b

3

-

194b

4

262d

209d

5

318b

-

6

370c

523c

7

395b

-

  • The authors should have connected the exhaust gas line of the thermogravimetric equipment on-line with a mass spectrometer to try to find out which compounds are released during the TGA.
  • Response:

We agree with the Reviewer on the point of connected the exhaust gas line to a mass spectrometer, which would provide more information on the gaseous products. However, we could not show that due to our limited analytical capability.

  • In section 3.2.1 it is said that the addition of the dispersed catalysts increases the liquid product fractions and decreases the coke formation. However, two lines below it is shown that the yield of liquids with these catalysts is lower than that obtained by thermal cracking.
  • Response:

We thank the Reviewer for pointing this out. We have elaborated the discussion in the revised manuscript in line 292-295 to be as follows:

“… The use of Ni-TBC[6] and Co-TBC[6] as standalone dispersed catalysts produced total liquid yields of 28.4 wt.% and 28.8 wt.%, which are comparable to that reported for the thermal run (29.9 wt.%). Nevertheless, mixing the Ni-TBC[6] and Co-TBC[6] were found to have the highest naphtha and distillate yields compared to the sole dispersed catalysts with a total liquid yields of ~32 wt.%.”

  • Figures 5 and 6 should also show the yield to coke or at least say that it is negligible.
  • Response:

We agree with the Reviewer that coke yield should be mentioned although it was negligible. Nevertheless, we added clarification of this point at the end of captions of Figures 5 and 7 in the revised version of the manuscript as follows:

“… Coke yield for all cases was negligible.”

  • In section 3.2.2 it is said that the properties of the supported commercial catalyst are shown in Table S4. However in that Table S4, the properties of the VGO are shown. That is, in the supplementary material the properties of that commercial catalyst are not shown.
  • Response:

We thank the Reviewer for this comment. We added a table for the Properties of the Commercial Hydrocracking Catalyst used in this study in the supplementary material as follows:

Table S5 Properties of the Commercial Hydrocracking Catalyst (KC-2710).

Property

Unit

Value

BET specific surface area

m2/g

346

Specific pore volume

mL/g

0.37

Average pore diameter

nm

4.3

Specific total acidity

µmol/g

844

Chemical composition:

SiO2

wt.%

33

Al2O3

wt.%

38

WO3

wt.%

23

NiO

wt.%

6

Support phase

amorphous SiO2-Al2O3 and Y zeolite (45 wt.%)

  • Perhaps, the most interesting thing would have been to analyze the composition of the liquid fractions from thermal hydrocracking and catalytic hydrocracking to compare their composition and really see the usefulness of using these catalysts.
  • Response:

The heavy oil contains wide ranges of different hydrocarbon components that makes it very complex in terms of characterization. In this study we are after proving the promotional effects of implementing the metal-based calixarenes catalyst precursor in enhancing the catalytic hydrogenation. This is reflected in maximizing high-value liquid yields. Therefore, we adopted lumping technique based on distillation boiling ranges which is considered to be an adequate technique where it is widely used in literature.

Round 2

Reviewer 1 Report

Thank you